# Sec16 alternative splicing dynamically controls COPII transport efficiency

Ilka Wilhelmi[1,†], Regina Kanski[1], Alexander Neumann[1], Olga Herdt[1], Florian Hoff[2], Ralf Jacob[2], Marco Preußner[1] & Florian Heyd[1]

The transport of secretory proteins from the endoplasmic reticulum (ER) to the Golgi depends on COPII-coated vesicles. While the basic principles of the COPII machinery have been identified, it remains largely unknown how COPII transport is regulated to accommodate tissue- or activation-specific differences in cargo load and identity. Here we show that activation-induced alternative splicing of Sec16 controls adaptation of COPII transport to increased secretory cargo upon T-cell activation. Using splice-site blocking morpholinos and CRISPR/Cas9-mediated genome engineering, we show that the number of ER exit sites, COPII dynamics and transport efficiency depend on Sec16 alternative splicing. As the mechanistic basis, we suggest the C-terminal Sec16 domain to be a splicing-controlled protein interaction platform, with individual isoforms showing differential abilities to recruit COPII components. Our work connects the COPII pathway with alternative splicing, adding a new regulatory layer to protein secretion and its adaptation to changing cellular environments.

---

[1] Department of Biology, Chemistry, Pharmacy, Freie Universität Berlin, Institute of Chemistry and Biochemistry, Laboratory of RNA Biochemistry, Takustrasse 6, 14195 Berlin, Germany. [2] Department of Cell Biology and Cell Pathology, Philipps-University Marburg, Robert Koch Strae 6, 35037 Marburg, Germany. † Present addresses: Department of Experimental Diabetology, German Institute of Human Nutrition Potsdam-Rehbruecke, Arthur-Scheunert-Allee 114-116, D-14558 Nuthetal, Germany (I.W.). Correspondence and requests for materials should be addressed to F.He. (email: florian.heyd@fu-berlin.de).

The early secretory pathway, the transport from the endoplasmic reticulum (ER) to the Golgi, is initially mediated by COPII-coated vesicles[1]. The COPII coat consists of an inner and an outer layer that are made up of Sec23–Sec24 heterodimers and Sec13–Sec31 heterotetramers, respectively[2]. The formation of COPII-coated vesicles is initiated by the ER membrane located guanine-nucleotide-exchange factor Sec12, which activates the small GTPase Sar1. In the GTP-bound state, Sar1 is membrane-associated and recruits Sec23–24 to concentrate cargo and form a pre-budding complex. Binding of Sec13–31 then leads to cage formation and finally vesicle budding. Eventually, the GTPase-activating protein (GAP) activity of Sec23, which is stimulated by Sec31, leads to hydrolysis of the Sar1-bound GTP[2]. GTP hydrolysis has been suggested to control cargo sorting[3], coat disassembly[4] and vesicle release[5]. The latter has been called into question, as a recent study finds vesicle scission independent of GTP hydrolysis[6].

COPII vesicles form at specialized sites of the ER, the transitional ER (tER), more generally termed ER exit sites (ERESs)[7]. Sec16 is a peripheral membrane protein that localizes to and defines tER/ERES[8–11]. Although vesicle budding can be reconstituted in the absence of Sec16 in vitro[4], the loss of Sec16 in cells leads to disruption of the early secretory pathway and growth arrest[8,12,13]. In humans, a Sec16 paralogue, Sec16B, has been suggested to fulfil specialized, non-redundant functions[14,15]. Human Sec16 is a protein of 2,357 amino acids that displays a central conserved domain involved in ERES localization[10,13], as well as a C-terminal conserved region (CTR) (Fig. 1a). The CTR was shown to interact with several COPII components such as Sec12 and Sec23, and is essential for COPII vesicle formation[13,16] (also see below). In addition, Sec16 interacts with Sec13, Sec31 and Sar1, and was suggested to play a scaffolding role for COPII coat formation without becoming part of the budding vesicle itself[8–10,17]. Later reports also find a direct regulatory role of Sec16 as it interferes with the Sec31-mediated increase in the GAP activity of Sec23. This results in increased Sar1-GTP stability and may influence formation and turnover of the COPII coat[5,18–20].

Upon T-cell activation, the expression and secretion of effector molecules such as cytokines, chemokines and cytotoxins is strongly increased. Accordingly, T cells have evolved an elaborate system that allows regulated and directed secretion of effector proteins[21,22]. However, research aiming at understanding protein secretion upon T-cell activation almost exclusively focuses on post-Golgi compartments. In contrast, it remains elusive how the early secretory pathway copes with the increased secretory cargo load upon T-cell activation. In experimental settings, an increase of secretory cargo led to different adaptive mechanisms, possibly controlled by phosphorylation of Sec16 (refs 23–25); however, this has not been investigated in endogenous settings. Another currently discussed mechanism to regulate the COPII machinery is the expression of different variants of COPII components[26]. In humans, two paralogues of Sec16, Sar1, Sec23 and Sec31, respectively, and four Sec24 paralogues exist. These variants are expressed in a tissue-specific manner[27,28] and mutations in a single gene, for example, Sar1B or Sec23A, are associated with metabolic and developmental diseases[29]. While the role of different paralogues of COPII components is a subject of intense research[28], a potential role of alternative splicing in controlling functionality of individual members of the COPII machinery has not yet been addressed.

Alternative splicing is a mechanism that multiplies the genome's coding capacity and that has an enormous, yet largely unexplored, regulatory potential. In addition, signal-induced alternative splicing dynamically controls protein expression in different cell types under diverse conditions[30,31]. In recent work, RNA sequencing was used to identify exons that are alternatively spliced upon T-cell activation[32]. Among these are two exons in the C-terminal region of Sec16, exons 29 and 30 (Fig. 1). Here we show that an increased expression of the Sec16 isoform containing only exon 29 leads to an increase in the number of ERES and more efficient COPII transport in activated T cells, thus allowing an adaptation to higher secretory cargo flux. We furthermore show that the different Sec16 splice variants have altered abilities to interact with COPII components and that Sec16 exon 29 controls COPII dynamics. Together, our data suggest that the C-terminal domain of Sec16 represents a platform for protein–protein interactions that is controlled by alternative splicing to regulate COPII vesicle formation. By linking dynamic changes in alternative splicing to the efficiency of COPII transport, we add a new regulatory layer to the early secretory pathway and provide evidence for an adaptive mechanism to increased endogenous secretory cargo.

## Results

**Sec16 is alternatively spliced upon T-cell activation**. A recent RNA sequencing approach identified over 100 exons that show activation-induced alternative splicing upon activation of the Jurkat-derived human Jsl1 T-cell line[32,33]. Among the alternatively spliced exons are exons 29 and 30 of Sec16 (Fig. 1; ref. 32) that make up a part of the CTR of the protein (Fig. 1a, left site shows domain organization of the Sec16 protein, right site shows exons that make up the Sec16 CTR and main splicing isoforms found in Jsl1 T cells). We first used splicing-sensitive RT-PCR to confirm these results. These experiments show an increase of the isoform containing only exon 29 (E29) and a concomitant decrease in the full-length (Fl) and the exon 30 (E30) containing isoforms in activated T cells (Fig. 1b,c). We confirmed that changed isoform expression was due to a splicing switch and not due to selective stabilization by showing similar stabilities of the different messenger RNA (mRNA) isoforms in resting and activated conditions (Supplementary Fig. 1a). While we observe a switch in Sec16 isoform expression at the mRNA level, the overall protein expression remained constant after T-cell activation (Fig. 1d, left). In a standard minigel SDS–polyacrylamide gel electrophoresis (PAGE), we do not observe a change in the electrophoretic mobility of Sec16 protein, which runs as a single band under these conditions. This is likely due to the small size difference of the different isoforms, as the alternative exons account for only 1–2% of the total protein. However, using a 4–12% gradient gel, we do detect at least two Sec16 bands the ratio of which is altered upon T-cell activation (Fig. 1d, right, we interpret the slower migrating isoform increased upon activation as being the E29 isoform; also see below). In agreement with altered Sec16 isoform expression at the protein level 48 h post stimulation, we observe a substantial increase in the E29 mRNA isoform already 24 h after activation (Supplementary Fig. 1b). In addition, we used Cycloheximide to block de novo protein synthesis to determine Sec16 protein stability and turnover in resting and stimulated T cells. In both conditions, we observed a rather short Sec16 protein half-life of ~10 h (Supplementary Fig. 1c), which is in the same range as previously observed in HeLa cells[34]. Together, these observations strongly suggest that alternative splicing leads to altered Sec16 isoform expression at the protein level 48 h post stimulation. Therefore, this time point was chosen for further investigation.

**The Sec16 E29 CTR interferes with ER-to-Golgi transport**. The alternatively spliced exons make up parts of the C-terminal domain of Sec16 (Fig. 1a) that has been shown to directly interact with Sec12 and Sec23, and to be essential for the function of Sec16

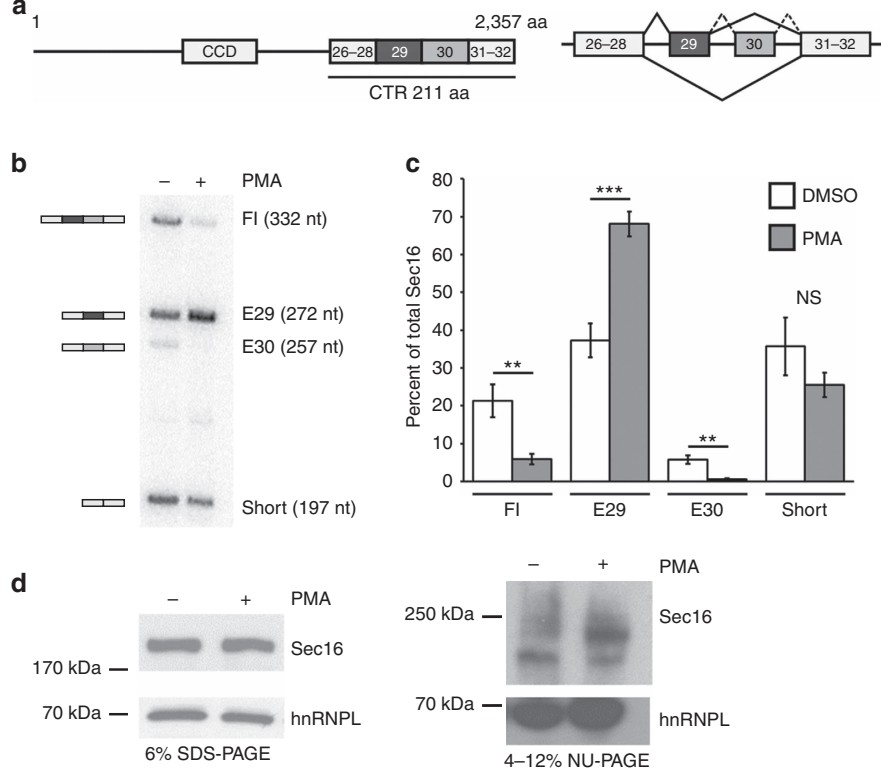

**Figure 1 | *Sec16* exons 29 and 30 are alternatively spliced on T-cell activation.** (**a**) Domain structure of the Sec16 protein (left) and schematic splicing pattern of the exons making up the CTR in Jsl1 T cells (right). CCD, central conserved domain; CTR, C-terminal region. The C-terminal region of Sec16 contains 211 amino acids in the isoform containing exons 26–32. Exons are not to scale. (**b**) Radioactive splicing-sensitive RT–PCR of resting ( − ) and stimulated ( + ) Jsl1 T cells detects four different splice isoforms. Schematic representation (left) and nomenclature used throughout the manuscript (right) of the four isoforms is shown. (**c**) Phosphorimager quantification of three independent experiments as shown in **b**. Shown is the mean amount of the individual splice isoforms as percentage of total *Sec16* ± s.d. *P* values (Student's *T*-Test) from left to right: 0,0041; 0,00067; 0,0015 and 0,099. (**d**) Western blot analysis comparing whole-cell extract of resting and stimulated Jsl1 T cells. Left panel shows separation on a standard 6% SDS–PAGE and right panel shows separation on a 4–12% gradient NuPAGE. hnRNPL served as a loading control.

to mediate COPII transport[13,16,18]. To elucidate potential functional differences of the four Sec16 isoforms, we over-expressed the different C-terminal domains in HeLa cells (Fig. 2a). When overexpressed, the C-terminal Sec16 domain is localized throughout the cytoplasm and disrupts ER-to-Golgi transport leading to Golgi breakdown, likely due to sequestering COPII components (Fig. 2b; ref. 13). Overexpressing similar amounts of the different Sec16 C-termini (Fig. 2c) revealed a particularly strong ability of the E29 isoform to interfere with ER-to-Golgi transport as evidenced by quantifying Golgi morphology (Fig. 2d; Supplementary Fig. 2). This could indicate an increased interaction of the E29 isoform with one of the COPII components, leading to the stronger dominant-negative effect. Conversely, it suggests that endogenous expression levels of the complete Sec16 E29 isoform, which is upregulated on T-cell activation, may regulate the efficiency of COPII transport.

**T-cell activation increases ERES and ER export efficiency.** We hypothesized that *Sec16* alternative splicing could be involved in an adaptation of the COPII machinery during T-cell activation. Although an adaptive mechanism to accommodate the strong increase in secretory cargo in activated T cells seems required, if and how such an adaptation takes place is unknown. To investigate this question and a potential connection to *Sec16* alternative splicing, we first compared the ER export capacity in resting and activated T cells using a fluorescence-activated cell sorting (FACS)-based modified vesicular-stomatitis-virus-

glycoprotein (VSVG) export assay (Fig. 3a, left). To this end, a FLAG epitope was inserted into the extracellular domain of the VSVG-GFP reporter, which accumulates in the ER upon heat shock at 40 °C and is released at 32 °C (ref. 35). In intact cells, staining of the FLAG-tagged protein will only be possible, if VSVG was correctly exported after shifting the cells to the permissive temperature. We indeed observed more efficient VSVG export in stimulated cells (Fig. 3a, right), confirming increased secretory capacity in activated T cells. In addition, we generated Jsl1 T cells stably expressing the VSVG-GFP reporter and performed export assays using immunofluorescence (IF) as read-out (also see Fig. 5 and Methods for details). Culturing these T cells at 40 °C overnight and then switching the cells to 32 °C confirmed strongly increased ER export efficiency upon T-cell activation (Supplementary Fig. 3a–c).

Next, we assessed the morphology of compartments involved in the early secretory pathway. The ER (visualized by transfection with dsRed fused to an ER retention signal), tER (Sec16, see Discussion below and ref. 10), ERGIC (ERGIC53) and the Golgi (GM130) did not show differences between resting and activated T cells (Fig. 3b; Supplementary Fig. 4a). However, we did notice a substantial and significant increase in the number of COPII-positive structures as evidenced by staining for Sec31 (Fig. 3b,c). In addition, the localization of COPII structures changed, with staining in the nuclear periphery becoming apparent only in stimulated T cells. To confirm this finding with an independent COPII marker and to rule out a potential phorbol myristate acetate (PMA)-induced formation of Sec31 aggregates,

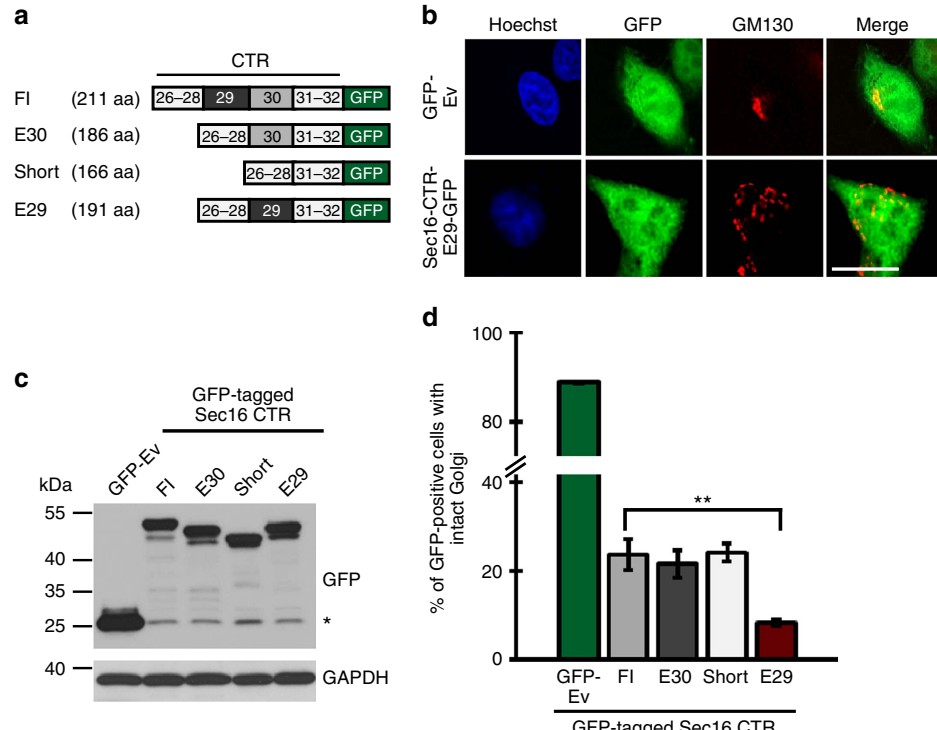

**Figure 2 | Different Sec16 isoforms interfere differentially with COPII transport.** (**a**) Schematic representation of the Sec16 CTR constructs fused to GFP. Exons are not to scale. (**b**) Expression of the Sec16 CTR leads to Golgi disruption. Immunofluorescence pictures of HeLa cells transfected and stained as indicated. Transfected cells were identified via GFP fluorescence and Golgi morphology was analysed using GM130 labelling. Control cells (top row) were transfected with a GFP empty vector (Ev) and show intact Golgi morphology, whereas cells overexpressing the GFP-tagged E29-CTR show a dispersed structure. (**c**) Western blot analysis of HeLa whole-cell extracts expressing the indicated constructs. GAPDH served as a loading control. * marks GFP alone. (**d**) The Sec16 E29 CTR leads to the strongest Golgi disruption. HeLa cells were transfected with either GFP control (Ev) or Sec16 CTRs fused to GFP. Golgi was labelled using GM130 antibody and morphology was analysed by fluorescence microscopy. At least 100 GFP-expressing cells were analysed in each of three independent experiments for each condition and evaluated for Golgi morphology. See Supplementary Fig. 2 for additional examples. Shown is the average percentage of counted cells showing an intact Golgi structure. $P = 0.0015$ (Student's $T$-Test).

we repeated the experiments with an antibody against Sec24C (ref. 36). As for Sec31, we observed a significant increase in the number of Sec24-positive structures upon PMA treatment (Supplementary Fig. 5), strongly suggesting that COPII-positive ERES or vesicles (see below) are indeed increased in activated T cells. The stronger effect we observe for Sec24C compared with Sec31 staining may be due to partial uncoating that mainly affects the outer COPII layer (Sec13/31), as the inner layer (Sec23/24) has been suggested to stay associated with vesicles after budding[37]. Notably, the activation of Jsl1 T cells with PMA did not induce ER stress, as the expression of two markers for ER stress remained unchanged between resting and activated conditions (Supplementary Fig. 6a). In addition, inducing ER stress did not lead to altered Sec16 alternative splicing (Supplementary Fig. 6b,c). We can thus rule out that the effect on *Sec16* alternative splicing and the COPII machinery is merely due to ER stress, suggesting an effect that is indeed due to T-cell activation (see below for the stimulation of primary cells). The majority of Sec24/31-positive structures, which do not costain with Sec16, has been suggested to represent COPII-coated vesicles with some nascent coats still associated with the ER stained as well[10,38]. This is consistent with our observation that Sec16- and Sec31-positive structures stain close to one another, but do not show a perfect co-localization (Fig. 3b; refs 10,39). Furthermore, Sec31-positive structures increase upon T-cell activation, but Sec16-positive tER does not (Fig. 3b,c), again suggesting that Sec16 and Sec31 stain (partially) non-overlapping structures. We thus consider it likely that we observe an increase in

COPII-coated vesicles in activated T cells; we will, however, use the more general term ERES to describe Sec31-positive structures in the following.

These data raise the possibility that the efficiency of COPII coat recruitment/formation increases upon T-cell activation, as we observe constant Sec16 (tER) staining, but an increase in dot-like COPII staining without a substantial change in overall protein levels (Supplementary Fig. 4b). Furthermore, our data support a model, in which an increase in ERES allows T cells to adapt their early secretory pathway to increased secretory cargo in activated T cells.

**Sec16 E29 controls ERES and ER export upon T-cell activation.** Having established an increase in the *Sec16* E29 isoform upon T-cell activation (Fig. 1), a role of this isoform in controlling the efficiency of ER-to-Golgi transport (Fig. 2) and an increase in ERES and ER export efficiency in activated T cells (Fig. 3), we went on to provide a direct link between these observations. Transfection of a splice-site blocking morpholino (MO) against exon 29 induced efficient exon exclusion (Fig. 4a–c). In fact, the increase of the *Sec16* E29 isoform upon activation was completely blocked in these cells, and the E29 level in stimulated E29MO-treated cells was reduced to that observed in unstimulated control cells (Fig. 4a–c). While we observe a clear change in Sec16 isoform expression upon E29MO transfection at the RNA level, the overall Sec16 protein level was unchanged (Fig. 4d, upper panel). Using a 4–12% gradient gel, we observed a

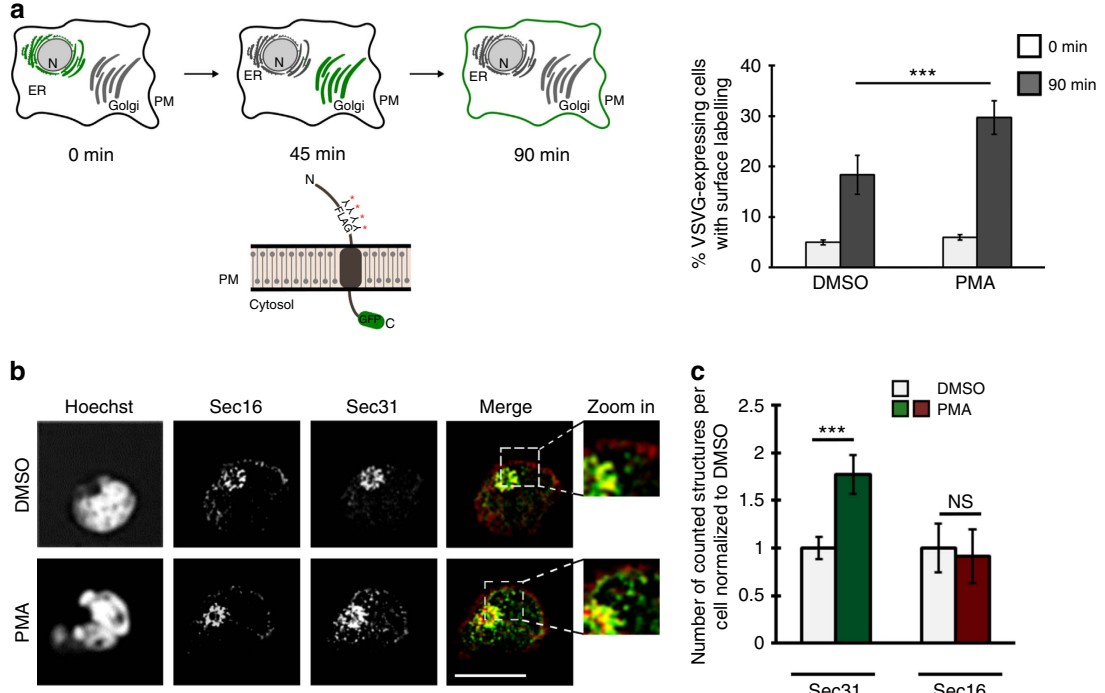

**Figure 3 | The number of ERES and transport efficiency increase upon T-cell activation.** (**a**) Schematic representation of a FACS-based export assay (left). The VSVG reporter remains in the ER upon 40 °C heat shock and is secreted after shifting the cells to 32 °C. Localization of the reporter at the different time points is indicated by green labelling of the corresponding cellular compartment. For surface labelling, a FLAG coding sequence was inserted in frame into the extracellular domain of the reporter (left, bottom). This assay was used to quantify ER export in resting (DMSO) and activated (PMA) Jsl1 T cells (right). Shown is the mean percentage of VSVG-expressing cells that showed FLAG surface labelling at the indicated time points. $N = 4$, error bars represent s.d. $P$ value is $6.5 \times 10^{-6}$. See text and Methods for details. (**b**) Activated T cells show constant tER staining but an increase in ERES number. Fluorescence pictures of resting (DMSO) and stimulated (PMA) Jsl1 T cells. Cells were stained for tER (Sec16/red) and ERES (Sec31/green). (**c**) Cells as in **b** were quantified using the ImageJ particle counting tool as described in Methods. Shown is the average number of counted structures normalized to DMSO ± s.d. of three independent experiments including 15 cells each. Raw numbers for Sec31 DMSO: 39.3 ± 4.5; Sec31 PMA: 68.9 ± 3.7; Sec16 DMSO: 33.8 ± 8.6; Sec16 PMA: 29.3 ± 4.7. $P = 0.00094$ (Student's $T$-Test) for Sec31 and $P = 0.47$ (Student's $T$-Test) for Sec16. DMSO, dimethylsulphoxide.

slower migrating isoform as the main Sec16 isoform upon PMA treatment in CoMO-transfected cells. In contrast, the switch to the slower migrating Sec16 isoform was markedly reduced in E29MO-transfected cells, confirming the effect of the E29MO on the protein level (Fig. 4d lower panel). On the basis of these results, we interpret the slower migrating band as the exon 29 containing isoform and the faster migrating band to represent the short isoform lacking both exons; this is consistent with these isoforms representing the main isoforms on the RNA level (Fig. 1). Furthermore, as the Sec16 isoform ratio can be altered by a specific MO, the different bands are most likely not the result of post-translational modifications. Stimulation of the CoMO-transfected cells led to the expected increase in ERES, while, as in untransfected cells, the tER was unchanged (Fig. 4e,f). In contrast, transfection of the cells with the E29MO completely abolished the PMA-induced increase in ERES, whereas the MO had no influence on the tER (Fig. 4e,f). Our data showing that the E29MO reverses the increase in ERES upon PMA stimulation provide direct evidence that *Sec16* alternative splicing, likely the increase in the E29 isoform, mediates this process. To further validate this hypothesis, we used a MO against *Sec16* exon 30 and investigated the effect on Sec16 isoform expression and ERES number upon T-cell activation. The E30MO reduced the Fl and the E30 isoform, and led to a slight increase in the short *Sec16* isoform (Supplementary Fig. 7a). ERES number in these cells was only slightly reduced compared with CoMO-treated cells and this reduction was not statistically significant. Similarly, combining both MOs did not increase the effect on ERES number observed

by transfecting the E29MO alone (Supplementary Fig. 7b). These data suggest that it is neither the Fl nor the E30 isoform that play the main role in controlling ERES number. Furthermore, the E30MO led to an increase of the short isoform without resulting in a significant change in ERES number. While this is consistent with the E29 isoform playing the predominant role in this process, we cannot rule out that parts of the effect observed in E29MO-treated cells is due to the strong increase of the short *Sec16* isoform (also see Discussion).

To directly connect *Sec16* alternative splicing with ER export efficiency upon T-cell activation, we performed the FACS-based ER export assay introduced above. As before, CoMO-transfected cells showed increased export efficiency upon PMA stimulation (Fig. 4g). Importantly, this PMA-induced increase in export efficiency was completely abolished in cells that were transfected with the E29MO (Fig. 4g). These data strongly suggest that Sec16 alternative splicing, by increasing ERES number (likely COPII-coated vesicles, see above), allows an adaptation of the early secretory pathway to higher cargo flux upon T-cell activation.

**Sec16 exon 29 regulates the efficiency of ER export.** To confirm an effect of *Sec16* alternative splicing on ER export efficiency in a more general system, we turned to HeLa cells. All four Sec16 isoforms were expressed in HeLa cells, and as in Jsl1 T cells, transfecting the E29MO resulted in a considerable decrease in the isoforms containing exon 29 (Fig. 5a,b). Consistent with the result in Jsl1 T cells, this was accompanied by a significant loss of ERES,

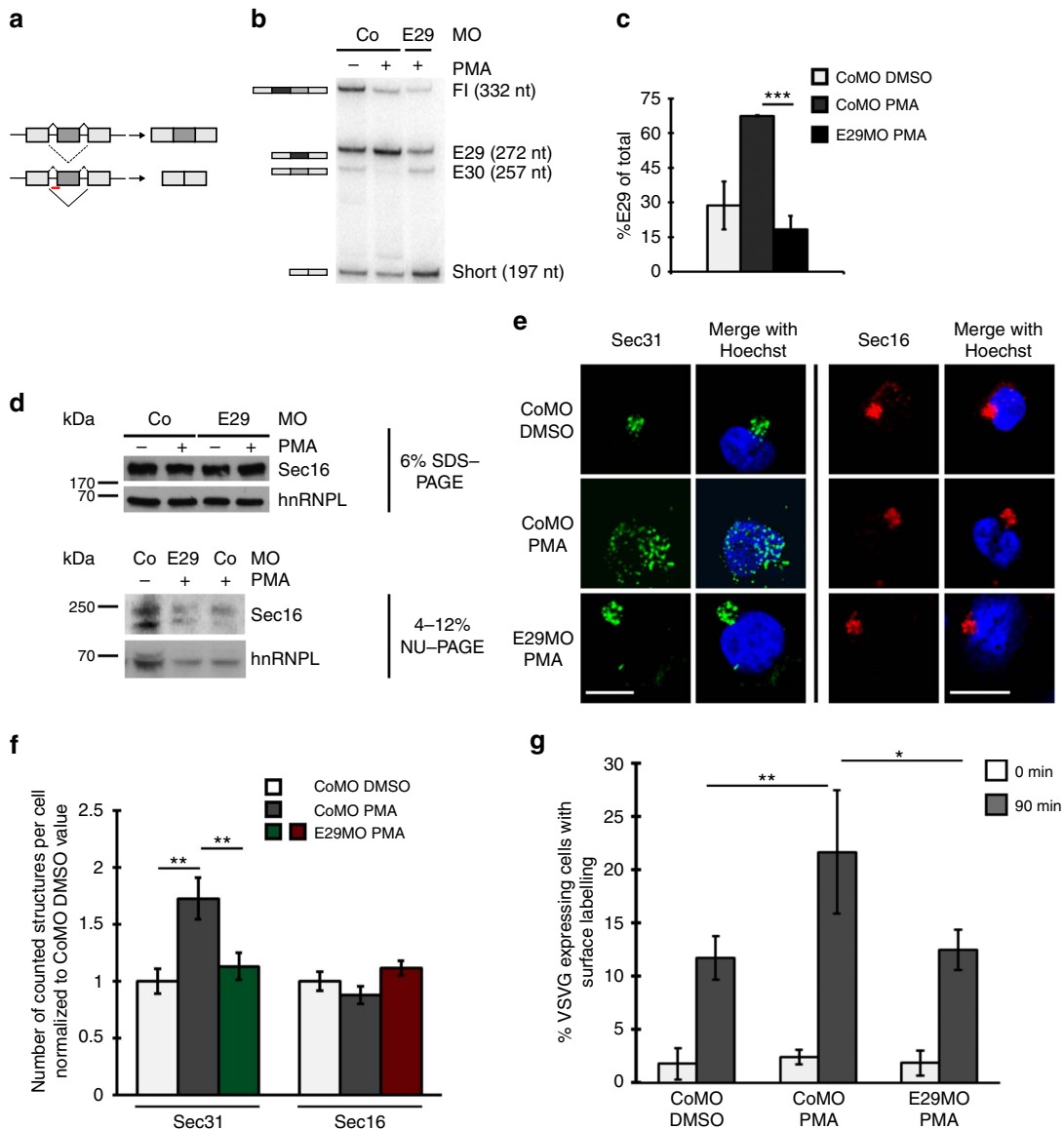

**Figure 4 | *Sec16* alternative splicing is required for the increase in ERES upon T-cell activation.** (**a**) Schematic view of the morpholino (MO) experiment. The exon in dark grey represents the exon targeted by the MO illustrated by the red bar. (**b**) Radioactive splicing-sensitive RT–PCR as in Fig. 1b of resting and stimulated Jsl1 T cells transfected with MOs as indicated and confirming a specific effect of the E29MO. (**c**) Phosphorimager quantification of the E29 isoform of three independent experiments as in **b**. Shown is the mean amount of the E29 isoform as percentage of total *Sec16* ± s.d. P value = $1.3 \times 10^{-4}$. (**d**) Total Sec16 protein level is not changed in MO-treated cells. Western blot analysis of whole-cell extract prepared from resting and stimulated Jsl1 cells transfected with either a control MO (CoMO) or a MO targeting Sec16 exon 29 (E29MO). Upper panel shows separation on a standard 6% SDS–PAGE and lower panel shows separation on a 4–12% gradient NuPAGE. hnRNPL served as loading control. (**e**) E29MO treatment prevents the increase of ERES on T-cell activation. Immunofluorescence pictures comparing Sec31 (green) and Sec16 (red) in MO-transfected cells ± PMA. Scale bar, 20 μm. (**f**) Cells described in **e** were analysed using the ImageJ particle counting tool (see Methods for details). Shown is the average number of counted structures normalized to CoMO DMSO ± s.d. of three independent experiments with 15 cells each. Raw numbers for Sec31: CoMO DMSO: 35.5 ± 3.9; CoMO PMA: 61.4 ± 6.5; E29 PMA: 40.2 ± 4.2. Raw numbers for Sec16: CoMO DMSO: 39.4 ± 3.2; CoMO PMA: 34.6 ± 3.1; E29 PMA: 40.4 ± 2.6. P values are 0.00409 (CoMO DMSO-PMA) and 0.00912 (PMA CoMO-E29MO). (**g**) FACS-based export assay using FLAG-VSVG-GFP as described in Methods. Shown is the percentage of VSVG-expressing cells with FLAG surface labelling after the indicated times at 32 °C under the conditions indicated. N = 3, error bar represents s.d. P values are: 0.004 and 0.048 (Student's *T*-Test). DMSO, dimethylsulphoxide.

whereas the tER remained unchanged (Fig. 5c,d). Notably, the magnitude of decrease in ERES upon E29MO treatment was similar to the effect observed by short interfering RNA (siRNA)-mediated knockdown of complete Sec16 in HeLa cells, which reduced ERES to 60–70% of control[23,34].

To confirm a direct influence of Sec16 isoform expression in controlling the efficiency of ER-to-Golgi transport, we used a VSVG-based export assay[35]. HeLa cells were sequentially transfected with the Sec16 E29MO and a VSVG-GFP reporter. Cells were then incubated at 40 °C overnight to retain VSVG-GFP in the ER, which was released by shifting the cells to 32 °C. We then quantified the distribution of VSVG-GFP in the ER or post-ER compartments at different time points upon shifting to the permissive temperature. In this assay, the MO-mediated decrease of exon 29 significantly reduced the efficiency of ER-to-Golgi transport (see Fig. 5e and Supplementary Fig. 8 for

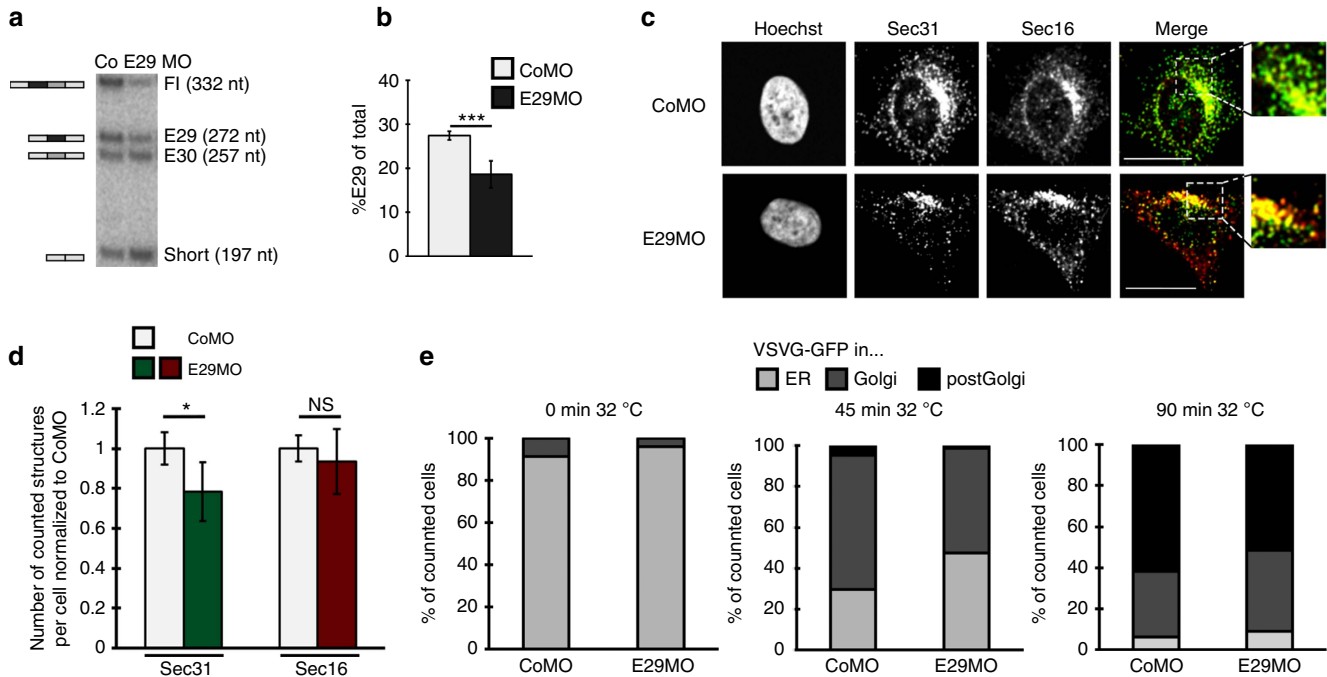

**Figure 5 | Sec16 alternative splicing regulates the efficiency of ER-to-Golgi transport in HeLa cells.** (**a**) The Sec16 E29MO works efficiently in HeLa cells. Radioactive splicing-sensitive RT–PCR as in Fig. 1b of HeLa cells transfected with either CoMO or Sec16 E29MO. (**b**) Phosphorimager analysis of four independent experiments described in **a**. Shown is the mean amount of the E29 isoform as percentage of total Sec16 ± s.d. P value $2.6 \times 10^{-5}$. (**c**) Sec16 E29MO-treated cells show reduced ERES number and constant tER staining. Immunofluorescence pictures of HeLa cells transfected with either CoMO (top) or E29MO (bottom) stained for Sec16 (red) and Sec31 (green). Scale bar, 20 µm. (**d**) Cells as in **c** were analysed using the ImageJ particle counting tool. Shown is the mean number of counted structures of three independent experiments with 15 cells each normalized to CoMO cells ± s.d. Raw numbers for Sec31: CoMO: 183.7 ± 14.9; E29MO: 142.2 ± 21.7; P = 0.037. For Sec16: CoMO: 130.8 ± 8.7; E29MO: 122.7 ± 25.6. (**e**) The Sec16 E29MO reduces ER export efficiency. VSVG-based export assay in MO-treated HeLa cells as described in Methods. At least 50 cells per condition were analysed per experiment in three independent experiments, shown is the average. Left (0 min): CoMO: cells with VSVG in: ER: 91.1 ± 2.8%; in Golgi: 8.9 ± 2.8%; E29MO: ER: 95.8 ± 1.3%; Golgi: 4.2 ± 1.3%. Middle (45 min): CoMO: ER: 29.8 ± 3.6%; Golgi: 65.7 ± 5.5%; post-Golgi: 4.5 ± 3.3%; E29MO: ER: 47.6 ± 3.7%; Golgi: 51.3 ± 2.7%; post-Golgi: 1.1 ± 1.1%; right (90 min): CoMO: ER: 6.1 ± 2.6%; Golgi: 31.9 ± 1.9%; post-Golgi: 61.9 ± 4.2%; E29MO: ER: 9.1 ± 3.5%; Golgi: 39.3 ± 0.6%; post-Golgi: 51.6 ± 2.9%. P values are P = 0.016 (Student's T-Test) for VSVG in Golgi after 45 min and P = 0.024 (Student's T-Test) for VSVG in post-Golgi structures after 90 min. For clarity, size marker have been omitted. Sar1A and B-GFP run around 50 kDa below the heavy chain; cytosolic Sec12-GFP runs between 75 and 100 kDa; Sec23-GFP has a predicted size of around 100 kDa and migrates with the corresponding marker band, the four Sec16-FLAG-tagged-CTRs migrate between the 25 and 15 kDa.

costaining of VSVG-GFP with ER and Golgi markers). As for the decrease in ERES number, the decrease in transport efficiency after E29MO treatment was comparable to the effect of siRNA-mediated Sec16 knockdown[8]. These data directly confirm a regulatory role of Sec16 isoform expression in the early secretory pathway and again suggest a particularly important role for Sec16 exon 29 in this regulation.

**Altered interaction of Sec16 isoforms with COPII components.** To provide a mechanistic basis for our observations, we performed immunoprecipitations (IPs) with the different Sec16 isoforms. We used the four FLAG-tagged Sec16 C-terminal constructs that were expressed at equal levels (Fig. 6a, input). After FLAG-IP, we investigated an interaction with GFP-tagged Sar1A/B, Sec12 and Sec23 as the COPII components that were suggested to directly interact with the C-terminal part of Sec16. Interestingly, exons 29 and 30 seem to differentially influence the interaction with different COPII components: both exons are required for efficient interaction with Sec12, exon 29 alone is sufficient to interact with Sec23 and the interaction with Sar1 requires exon 30 (Fig. 6a). As the IPs with Sec23 suggested the strongest interaction with the E29 isoform, we aimed to test the functional importance of this interaction. To this end, we

coexpressed the Sec16 E29 C terminus along with Sec23 and analysed Golgi disruption as in Fig. 2. While expression of Sec23 alone had no effect on Golgi morphology (Supplementary Fig. 9), the effect of the Sec16 E29 C terminus was completely rescued by coexpression of Sec23 (Fig. 6b,c). This indicates that the Sec16–Sec23 interaction is of crucial importance for the Sec16-mediated control of the early secretory pathway. Together, these experiments suggest a model, in which higher expression of the Sec16 E29 isoform and its increased interaction with Sec23 increase the efficiency of COPII transport in activated T cells.

**Sec16 E29 controls protein export in primary human T cells.** To investigate the physiological relevance of Sec16 exon 29, we analysed alternative splicing and its role in controlling protein secretion in primary human T cells. We used peripheral blood mononuclear cells (PBMCs) that were enriched for T cells and stimulated these cells with physiologically relevant stimuli, either α-CD3 or α-CD3/α-CD28. In both cases, we observed an increase in the exon 29 containing isoform and a concomitant decrease in the shortest isoform (Fig. 7a,b). This is consistent with the recent finding that PHA stimulation of primary human CD4+ T cells increases Sec16 exon 29 inclusion[32]. To test the functional impact of exon 29 in primary cells, we used MO-mediated exon exclusion

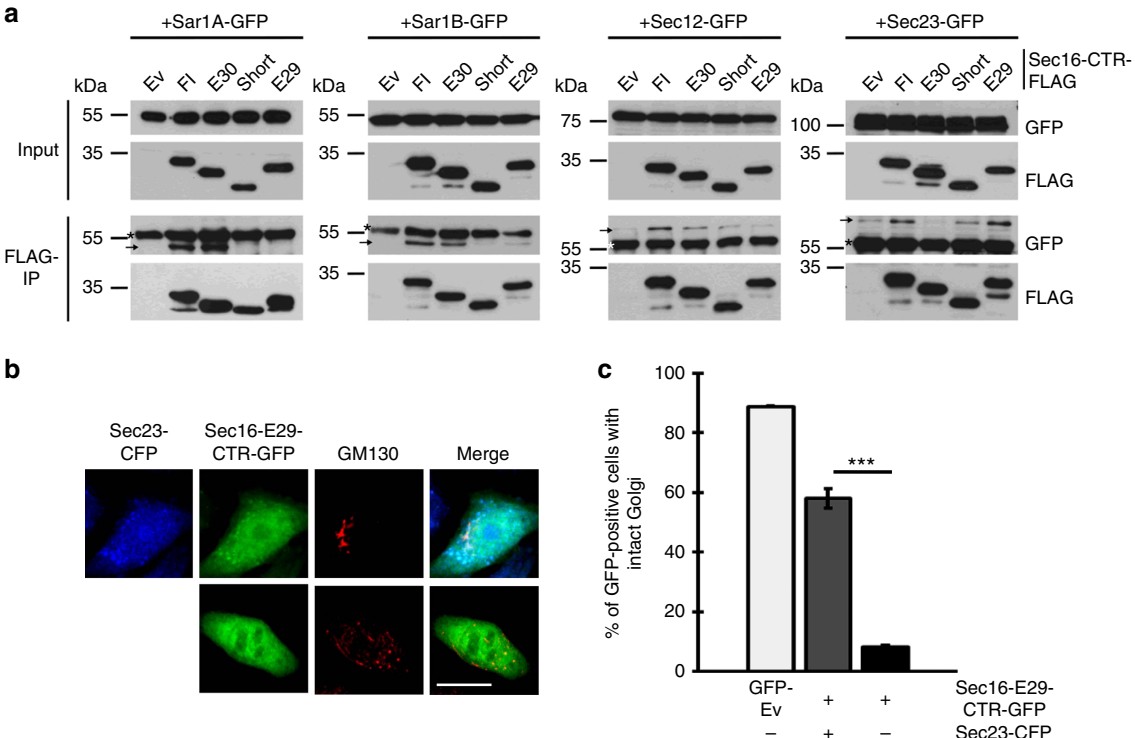

**Figure 6 | Different Sec16 isoforms interact differentially with COPII components.** (**a**) Coprecipitations of Sec16 isoforms with Sar1A, Sar1B, Sec12 and Sec23A show differential interactions. Hek293 cells were cotransfected with Sec16 CTRs fused to FLAG or FLAG-empty vector as control and COPII components fused to GFP. FLAG precipitates were analysed by western blot using FLAG and GFP antibodies. Top panels show 5% input and bottom panels show FLAG-IPs. → marks the coprecipitated protein and * marks the heavy chain. (**b**) Coexpressing Sec23 with the Sec16-E29-CTR-GFP rescues the effect on Golgi disruption. Fluorescence pictures of HeLa cells coexpressing Sec16-E29-CTR-GFP and Sec23A-CFP (top row) or Sec16-E29-CTR-GFP alone. Golgi is stained using GM130 antibody (red). Scale bar, 20 μm. (**c**) Cells as in **b** were analysed in three independent experiments with an average of 120 cells counted per experiment and compared with cells in Fig. 2c ($P = 0.00014$ (Student's T-Test)).

as in Fig. 4. Transfecting T-cell-enriched PBMCs with a Sec16 exon 29 splice-site blocking vivo-MO led to efficient exon exclusion as shown by RT–PCR (Fig. 7c). We used these cells, along with control vivo-MO-transfected cells, stimulated with α-CD3/α-CD28 and investigated secretion of interleukin (IL) 2 by enzyme-linked immunosorbent assay (ELISA). In this experiment, we observed a significantly reduced IL2 secretion in cells transfected with the E29MO (Fig. 7d). However, the amount of intracellular IL2 that was measured after inhibition of protein secretion was similar in E29MO- and CoMO-transfected cells, suggesting that Sec16 exon 29 exclusion does not alter total IL2 levels, but interferes with protein secretion (Fig. 7e). These data confirm our model in primary human T cells using a physiologically relevant stimulus. Altered export of an endogenous interleukin upon skipping of exon 29 emphasizes the relevance of Sec16 splicing in the regulation of T-cell function.

**Cells lacking Sec16 E29 show decreased ERES and ER export.** Finally, we confirmed the crucial importance of Sec16 exon 29 in the early secretory pathway using CRISPR/Cas9-mediated deletion of this exon in Hek293 cells. We obtained cells with homozygous deletion of exon 29 as assessed by genomic PCR and RT–PCR (see Fig. 8a and Supplementary Fig. 10a for targeting strategy). RT–PCR confirmed that deletion of exon 29 abolished formation of the full-length and E29 isoforms, whereas the E30 and short isoforms were still expressed (Fig. 8a). Western blots showed that deletion of exon 29 did not change overall protein levels, suggesting that the lack of exon 29 led to the increased formation of remaining isoforms (Fig. 8b). In line with MO experiments, staining of exon 29-deleted cells

for Sec31 revealed a strongly reduced number and a more compact ERES localization when compared with wild-type (WT) Hek293 cells (Fig. 8c,d), while Sec16 staining itself (tER) showed no difference (Supplementary Fig. 10b). We then aimed to measure protein export in a VSVG-based export assay. However, as the exon 29-deleted cells showed reduced adherence, especially after the overnight incubation at 40 °C, we were unable to obtain enough cells for quantification after staining for IF. An EndoH assay qualitatively suggested a reduced efficiency of ER export in Hek293 cells lacking Sec16 exon 29 (Fig. 8e), but was difficult to quantify due to the proximity of the two VSVG species in the SDS gel. To quantitatively measure protein export we turned to the FLAG-modified VSVG and FACS as read-out as described above. These experiments confirmed a strong export defect in Sec16 exon 29-deleted cells, as the presence of the FLAG-tagged VSVG at the cell surface 90 min after shifting the cells to 32 °C was significantly reduced when compared to control cells (Fig. 8f).

The genome-engineered cells represent a well-suited model system to start addressing the mechanism of Sec16 exon 29 mediated control of export efficiency. Our data would be consistent with exon 29 mediating more efficient COPII coat formation, probably through increased interaction with Sec23. We therefore used Sec23 fluorescence recovery after photobleaching (FRAP) to compare COPII dynamics in WT and exon 29-deleted cells. As predicted by our model, cells lacking Sec16 exon 29 show a significantly increased Sec23 FRAP half-life (Fig. 8g), suggesting that COPII turnover is reduced in these cells. Together with the interaction studies in Fig. 6, these data suggest that Sec16 isoforms control

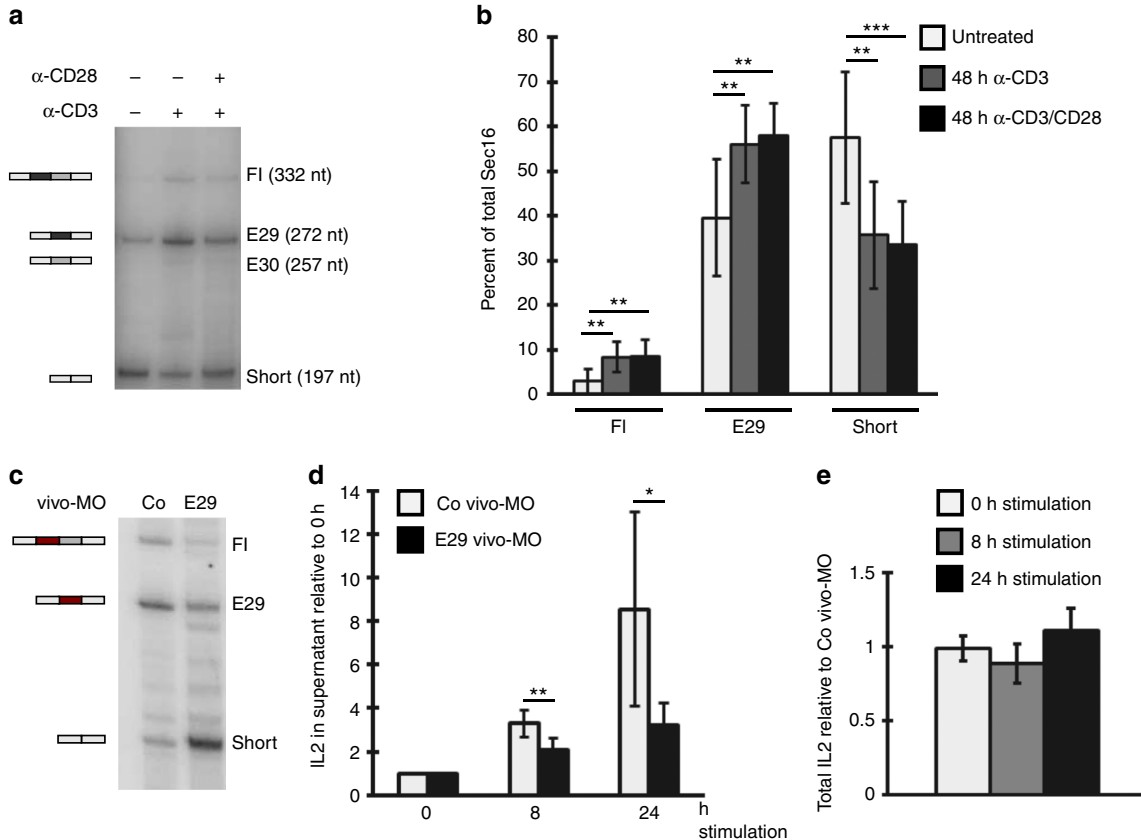

**Figure 7 | *Sec16* is alternatively spliced upon activation of primary human PBMCs. (a)** Primary human PBMCs were cultured and stimulated after lymphocyte enrichment (removing adherent cells) as described[33]. T-cell percentage was determined by FACS for three independent donors (66.9 ± 6.8% CD4 and 15.5 ± 4.7% CD8). RNA was prepared and *Sec16* isoform expression was analysed as in Fig. 1b. **(b)** Phosphorimager analysis shows the average percentage of *Sec16* isoforms. Shown is the average of four different donors measured in duplicates ± s.d. Note that E30 isoform is not reproducibly detectable in these primary cells. *P* values are for CD3: 0.002 (FI); 0.006 (E29); 0.003 (short); for CD3/28: 0.0029 (FI); 0.0018 (E29) and $8.7 \times 10^{-4}$ (short) (Student's *T*-Test). **(c)** Cells as in **a** were transfected with vivo-MOs and *Sec16* isoform expression was analysed as in **a**. **(d)** T-cell-enriched PBMCs as in **a** were transfected with vivo-MOs, stimulated for the indicated time with α-CD3/α-CD28 and IL2 in the supernatant was measured by ELISA at the indicated time points. Data are normalized to unstimulated cells ($t = 0$) and represent means of four independent experiments each with two independent stimulations and assessed in duplicate ± s.d. *P* values are: 0.0098 and 0.031 (Student's *T*-Test). **(e)** PBMCs stimulated as in **d** were additionally treated with BFA for the last 40 min, cells were then lysed and total IL2 was measured as in **d**. Shown is the amount of IL2 in E29MO-transfected cells relative to CoMO-transfected cells at each time point ± s.d. ($n = 4$).

COPII dynamics through differential interaction with COPII members to regulate the efficiency of the early secretory pathway (see model in Fig. 9).

## Discussion

Our data define the C terminus of Sec16 as a protein–protein interaction platform that is regulated by alternative splicing to control protein secretion upon T-cell activation. To our knowledge, this is the first example of an alternative splicing event that dynamically controls the early secretory pathway in response to changing cellular demands. Our data thus add a new regulatory layer to the control of COPII transport and describe an adaptive mechanism to increased endogenous secretory cargo.

Formation of COPII-coated vesicles has been investigated over the last decades, uncovering mechanistic and structural details. However, it remains largely unknown how the COPII machinery dynamically adapts to changes in secretory cargo load upon changing cellular conditions[34]. Similarly, it remains elusive how the COPII pathway adapts to tissue-specific differences in secretory cargo flux. The existence and tissue-specific expression of several paralogues of COPII components, for example, *Sar1A* and *Sar1B* or *Sec24A–D*[26] may be one mechanism to control the early

secretory pathway. This idea is supported by mouse models[28] and human diseases linked to mutations in only one of several paralogues[29]. However, functional differences between paralogues are only beginning to emerge[40] and the regulation of individual COPII members, for example by alternative splicing, is even less well understood. It is interesting to note that we have observed *Sec16* alternative splicing also in mouse and find tissue-specific splicing patterns (unpublished observation). This suggests that *Sec16* alternative splicing not only controls dynamic adaptation of the COPII machinery but also contributes to tissue-specific differences in the early secretory pathway, probably acting together with selective expression of COPII paralogues.

Upon T-cell activation, a resting cell strongly increases the expression of many secreted effector molecules such as interleukins, cytokines and cytotoxins. However, the adaptation of ER-to-Golgi transport to this increase in cargo flux has, until now, not been investigated. Our work shows that indeed export efficiency is strongly increased in activated T cells and suggests that the underlying mechanism is an increase in the number of ERES, more specifically COPII-coated vesicles, mediated by Sec16 alternative splicing. We thus provide evidence for an adaptive mechanism to increased endogenous secretory cargo, also in primary human cells. Notably, Sec16 has been identified as a protein controlling ER

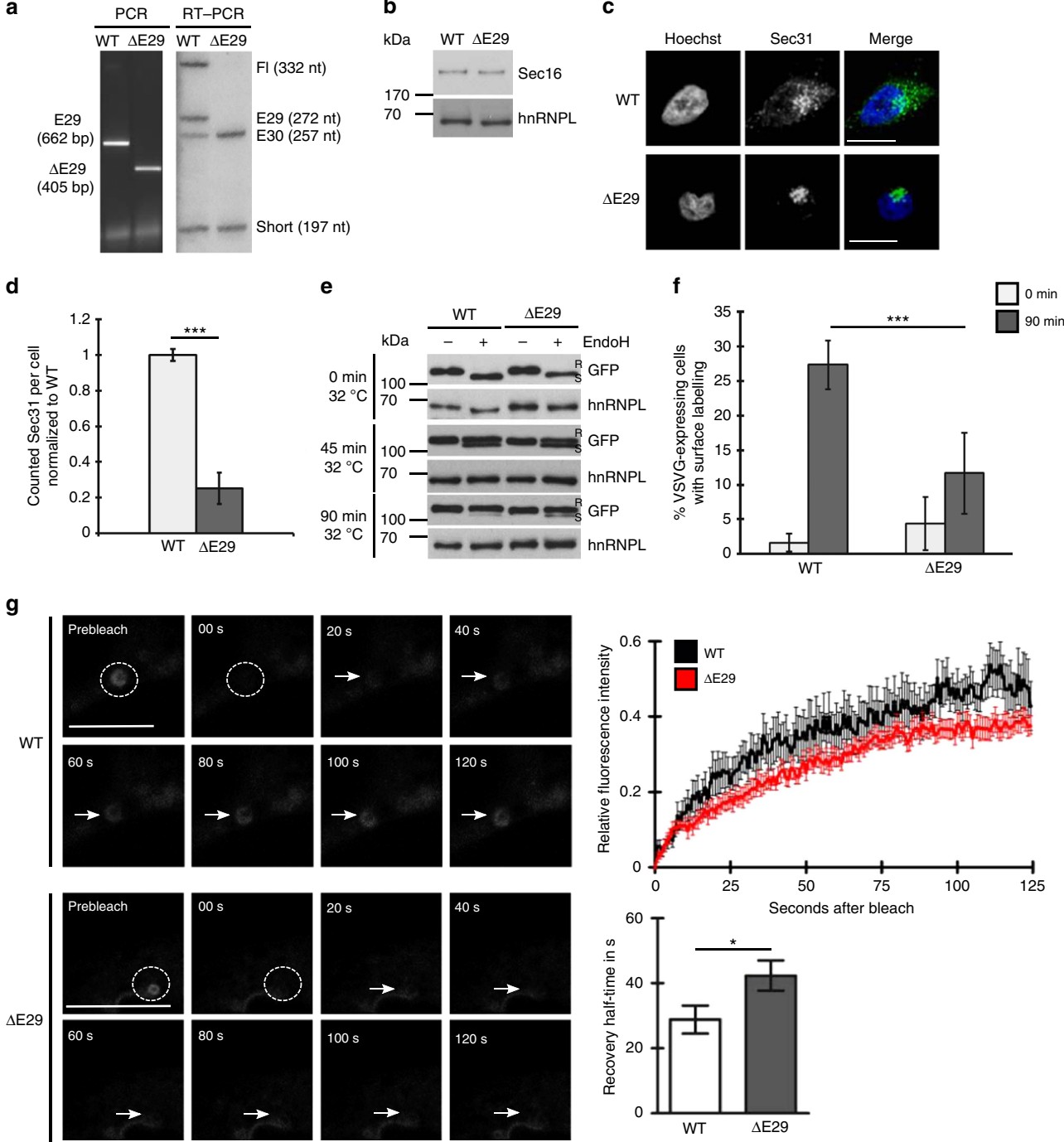

**Figure 8 | *Sec16* exon 29 is necessary for efficient ERES formation and ER-to-Golgi transport.** (**a**) CRISPR/Cas9-mediated deletion of *Sec16* exon 29 detected at the DNA and RNA level by genomic PCR (left) and RT–PCR (right). (**b**) Sec16 western blot of whole-cell extracts prepared from WT and ΔE29 Hek293 cell lines showing no difference of the total protein level. hnRNPL was used as a loading control. (**c**) Deletion of *Sec16* exon 29 strongly reduces ERES number. Immunofluorescence pictures of Hek293 WT and ΔE29 cells stained for Sec31. Scale bar, 20 μm. (**d**) Cells described in **c** were analysed using the ImageJ particle count tool. Shown is the mean number of counted Sec31-positive structures normalized to WT ± s.d. of three independent stainings with a total of 30 cells analysed. Raw numbers are for WT: 77.1 ± 2.6 for ΔE29: 19.5 ± 6.8. *P* = 0.0002 (Student's *T*-Test). (**e**) *Sec16* exon 29-deleted cells show a defect in ER export. Export assay performed in WT Hek293 and ΔE29 cells using the ts045-VSVG-GFP variant and EndoHf as described in Methods. VSVG was detected using GFP antibody. hnRNPL served as a loading control. R, resistant to EndoHf; S, sensitive to EndoHf. (**f**) Loss of *Sec16* exon 29 reduces protein secretion. Export assay performed in WT and ΔE29 Hek293 cells using a VSVG variant containing an internal FLAG-tag as a surface marker. Cells were stained with α-FLAG at the indicated time points after shifting the cells to 32 °C and secretion levels were measured via FACS. Shown is the average of three independent experiments. *P* = 0.00014. (**g**) COPII turnover rate is reduced in ΔE29 cells. WT and ΔE29 293Hek cells were transfected with a plasmid encoding Sec23 fused to a C-terminal GFP-tag. Sec23 turnover was analysed via FRAP. Left panel shows representative time laps pictures for each cell type. Scale bar, 10 μm. Upper right shows FRAP curves with slower recovery rate for Sec23 in ΔE29 cells and lower right shows a prolonged calculated half-life ± s.e.m. *n* = 15 from four independent experiments, *P* = 0.0124 (Student's *T*-Test).

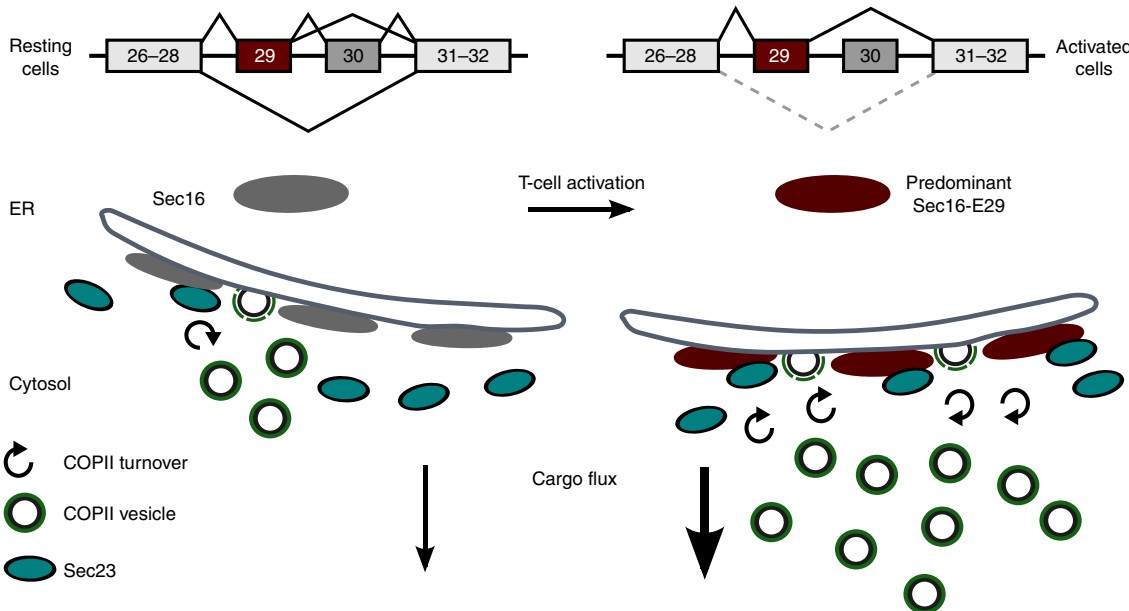

**Figure 9 | Model summarizing the adaptation of the early secretory pathway upon T-cell activation.** The main *Sec16* splice variants in resting and activated T cells are shown. The tER shape does not change upon T-cell activation, but Sec16 shows altered isoform expression (red). In activated T cells, Sec16 exon 29 containing isoforms show increased interaction with Sec23 and lead to an increase in COPII dynamics, COPII-positive structures (Sec31, green) and secretory cargo flux.

export in response to starvation[25], growth factor signalling and proliferation[24,34], and ectopic expression of secretory cargo[23]. In the latter study, a slight increase in the overall Sec16 expression level was observed[23] and changing Sec16 levels were also observed in response to growth factor signalling[34]. In addition, Sec16 phosphorylation has been shown to regulate its activity, for example by changing its intracellular localization in response to starvation[25]. These studies, together with our work, identify different levels of Sec16 modifications that, in response to changing cellular conditions, alter Sec16 functionality by distinct mechanisms. It is possible that Sec16 is modified by different mechanisms simultaneously, as we cannot rule out phosphorylation in addition to alternative splicing in activated T cells, and other studies have not investigated differential Sec16 isoform expression. Taken together, these studies show that several distinct signalling pathways converge on Sec16 to control ER export, suggesting Sec16 to play a central role in the dynamic adaptation of COPII transport in diverse cellular settings.

Our MO and CRISPR/Cas9 experiments clearly show an important role of *Sec16* alternative splicing, in particular exon 29, in regulating the COPII pathway. It is noteworthy that the MO-mediated change in *Sec16* alternative splicing has a similar effect on ERES number and export efficiency as siRNA-mediated knockdown of complete *Sec16* (refs 8,23,34). This is consistent with differential *Sec16* isoform expression playing a crucial role in controlling ERES formation and ER export. Our experimental conditions do lead to a reduction of the full-length and E29 isoform, and to an increase in the E30 and short isoforms with overall constant protein levels. These experiments do not distinguish whether it is the loss or gain of one of these isoforms that causes the observed phenotype. However, overexpression of the different C-termini and their effect on disrupting ER-to-Golgi transport provides additional information. In this experiment, the E29 C terminus clearly stands out as the isoform with a profound effect on the COPII pathway. These data together suggest that it is the loss of the

E29 isoform that causes the defect in MO-treated and genome-engineered cells. This conclusion is further supported by experiments using a MO targeting *Sec16* exon 30, which did not result in a significant change of ERES number despite altered *Sec16* isoform expression. In addition, in the endogenous situation it is the E29 isoform that changes most strongly upon T-cell activation, again pointing to this isoform to mediate functional changes. However, given the differential interaction of the different Sec16 isoforms with COPII components, it appears possible that expression of the other Sec16 splice variants and ultimately the isoform ratio contribute to the regulation of ER export efficiency in the endogenous situation. A model is conceivable in which different Sec16 isoforms preferentially recruit particular components of the COPII machinery, which could then influence the efficiency of COPII coat generation and vesicle budding. Unfortunately, overexpressing full-length Sec16 isoforms interfered with the early secretory pathway (unpublished observation and refs 8,13), preventing us from testing whether expression of the E29 isoform alone would lead to more ERES and more efficient ER export or whether additional isoforms need to be coexpressed.

Interestingly, the defect caused by overexpressing the E29 C terminus can be completely rescued by coexpressing Sec23. Sec23 appeared to interact stronger with the E29 than with the other isoforms and has been suggested to fulfil a central role in the COPII pathway[37]. Consistently, our data suggest a crucial role of the Sec16–Sec23 interaction in controlling COPII transport. It furthermore points to a model, in which efficient recruitment of Sec23 by the Sec16 E29 isoform, potentially in combination with altered interaction with other COPII components, increase the efficiency of COPII vesicle formation and ER-to-Golgi transport. This model is supported by our FRAP experiments showing reduced COPII dynamics in cells lacking *Sec16* exon 29. The precise molecular mechanism for this regulation is so far unknown. A mere scaffolding function of Sec16 is conceivable and a more efficient interaction of the E29 isoform with Sec23 could explain the observed effects. However, Sec16 has

been shown to negatively regulate Sar1 activity by interfering with the Sec31-mediated increase in the GAP activity of Sec23 (refs 5,18,19). It is thus possible that in addition to the scaffolding function, Sar1 GTPase activity is differentially regulated by Sec16 isoforms, which interact differentially with Sar1, Sec12 (GEF) and Sec23 (GAP). This could allow a fine-tuning of Sar1-GTP hydrolysis, potentially depending on the Sec16 isoform ratio, which could be a decisive factor in controlling COPII vesicle formation.

In addition to Sec16, several other ER-localized proteins with known functions in ER export were shown to be alternatively spliced upon T-cell activation[32]. We therefore expect that our present work is only the first example of a connection between alternative splicing and the early secretory pathway, which may turn out to be a common regulatory principle.

## Methods

**Cell culture and transfection.** Hek293 and HeLa cells were grown in DMEM supplemented with 10% fetal bovine serum and 1% penicillin/streptomycin (Invitrogen). Transfection of these cell lines was performed using Lipofectamine 2000 (Invitrogen) according to the manufacturer's instructions. For MO experiments, cells were seeded and transfected 1 day later using Endoporter following to the manufacturer's manual. MOs (Sec16 E29: ACCTGCCGAGGAAA AGAAGAATTGC and standard control) and Endoporter transfection reagent were purchased from Gene Tools.

Maintenance, transfection and stimulation of Jsl1 cells, and generation of stable cell lines were done essentially as described[32]. Stimulation was carried out using PMA (Sigma) at a final concentration of $20 \, ng \, ml^{-1}$ and an equal amount of dimethylsulfoxide as a vehicle control. Unless otherwise mentioned, experiments were performed 48 h post stimulation. For RNA stability assays, Jsl1 cells were treated with actinomycin D (Sigma) at a final concentration of $5 \, \mu g \, ml^{-1}$ for 0, 4 and 8 h, and cells were collected for RNA isolation. For the protein stability assay, cells were treated with cycloheximide at a final concentration of $40 \, \mu g \, ml^{-1}$ and cells were collected after 0, 6 and 10 h for further analysis.

For induction of ER stress, Jsl1 cells were treated with $1 \, \mu g \, ml^{-1}$ tunicamycin overnight. Stress induction was verified by quantitative RT–PCR (RT–qPCR) of marker genes.

For genome-engineering in Hek293 cells, intronic sequences flanking E29 of *Sec16* were analysed for single guide RNA (sgRNA) candidates *in silico*[41]. A pair of oligos for the highest ranked candidate sgRNAs (E29 5′-FW caccGAAGAGGGCG TTTGTAACCC, E29 5′Rev aaacGGGTTACAAACGCCCTCTTC, E29 3′-FW CaccgCTTTATGGAGGGGCGGTACT, E29 3′-Rev aaacAGTACCGCCCCTCCA TAAAGc) was synthesized and subcloned into the pSpCas9 (BB)-2A-Puro: PX459 vector (kindly provided by Stefan Mundlos). Hek293 cells were seeded on 24-well plates 24 h before transfection. Cells were transfected using Rotifect at 80–90% confluency following the manufacturer's protocol. For each well of a 24-well plate, a total of 500 ng Cas9 + sgRNA plasmid was used. Fourty-eight hours after transfection, the cells were treated with $1 \, \mu g \, ml^{-1}$ puromycin to select for cells expressing the sgRNAs. Genomic DNA was extracted using DNA extraction buffer (200 mM Tris (pH 8.3), 500 mM KCl, 5 mM $MgCl_2$ and 0.1% gelatine in $H_2O$) and a PCR was performed using gene-specific primers. First, the population of targeted cells was analysed and then clonal cell lines were isolated by dilution[41].

All cell lines were routinely tested and found negative for mycoplasma using a PCR-based method. Jsl1 cells were obtained from Kristen Lynch (UPenn), Hek293 and HeLa cells are lab stocks that have been extensively used over the past years and that show the expected morphological characteristics.

**Constructs.** ts045-VSVG-GFP was purchased from Addgene (#11912), GFP-Sec16 full-length expression plasmid was kindly provided by David Stephens (Bristol, UK). The Sec12 construct used for IPs corresponds to the cytoplasmic Sec12 used for a similar experiment before[16].

For all other expression constructs, open reading frames were PCR-amplified from complementary DNA using primers introducing restriction sites. Products were digested and ligated into pCMV-N3-FLAG or pCMV-N3-GFP expression vectors. All constructs were verified by sequencing.

For primer sequences see Supplementary Table 1.

**RNA and RT-PCR and RT-qPCR.** RNA was isolated using PeqGOLD RNApure (Peqlab) according to the manufacturer's instructions. RT–PCR was done as described[42]. Briefly, 1 μg RNA was used for RT–PCR using a gene-specific reverse primer for the RT reaction followed by a low-cycle PCR using a radioactive $^{32}P$-labelled forward primer. PCR products were separated on a denaturing PAGE. Quantification was done after phosphorimaging using ImageQuant TL software. Quantifications are shown as mean of independent experiments ± s.d. For Sec16 splice PCR, the following primer pair was used: forward: 5′-GATGGCCTCCTCGCTCTCTCATCCC-3′ and reverse: 5′-

GCCAGCTGAGCAGGGTTGTAGAAGG-3′. For RT–qPCRs, the RT protocol described above was used combining up to four gene-specific reverse primers. qPCR was performed using the ABsolute qPCR SYBR green mastermix in a 96-well-plate in a Stratagene MX3005p system.

**Western blotting and IP.** Whole-cell extracts (WCEs) were prepared with lysis buffer (60 mM Tris (pH7.5), 30 mM NaCl, 1 mM EDTA and 1% TritonX-100) supplemented with protease inhibitors. Concentrations were determined using Bradford test. SDS–PAGE, NuPAGE and western blotting were performed using standard protocols. For IPs, 50 μg WCEs was incubated in 500 μl modified RIPA buffer (10 mM Tris (pH8.0), 1% NP40, $5 \, mg \, ml^{-1}$ sodium deoxycholate, 2 mM EDTA and 200 mM NaCl) supplemented with protease inhibitors and pre-cleaned by rotating 1 h at 4 °C with protein-A/G sepharose beads (Santa Cruz). Pre-cleaned WCE was then supplemented with equilibrated FLAG-M2 beads (Sigma) and rotation was continued overnight at 4 °C. Beads were then washed five times and eluted by boiling 5 min in 2× SDS-loading buffer. Eluted samples were separated on SDS–PAGE and analysed by western blotting.

For all WBs where only the relevant part of the blot is shown in the main figure, full blots are presented in Supplementary Fig. 11.

**Antibodies.** The following primary antibodies were used: α-GFP (B2, sc9996, Santa Cruz, WB 1:500), α-FLAG (DYKDDDDK, 2368, Cell Signaling, WB 1:1,000), α-FLAG (M2, Sigma, FACS 1:400), α-hnRNPL (4D11, sc32317, Santa Cruz, WB loading control 1:5,000), α-GAPDH (GT239, GeneTex, WB loading control 1:5,000), α-Sec16 (KIAA0310, A300-648A-1, Bethyl laboratories, WB 1:1,000 and IF 1:200), α-Sec31A (sc-136233, Santa Cruz, WB 1:1,000 and IF 1:200), α-Sec24C (IF 1:100, kindly provided by David Stephens, ref. 36), α-GM130 (EP892Y, 1837-1, Epitomics, IF 1:400), α-ERGIC53 (E1031, Sigma, IF 1:400) and α-Sar1A (SAB1402607, Sigma, WB 1:500). The following conjugated secondary antibodies were used: α-mouse-HRP (7076S, Cell Signalling, WB 1:5,000), α-rabbit-HRP (7074S, Cell Signalling, WB 1:5,000), α-rabbit-Alexa647 (A21245, Life Technologies, IF 1:400), α-mouse-Alexa488 (A212002, Life Technologies, IF 1:400) and α-Mouse-Cy5 (Jackson Immuno Research, IF 1:400).

**Export assay and EndoHf.** Cells were transfected with the ts045-VSVG-GFP reporter construct and shifted to 40 °C overnight to retain the reporter protein in the ER. On the next day, cells were supplemented with cycloheximide at a final concentration of $100 \, \mu g \, ml^{-1}$ before they were shifted to the permissive temperature of 32 °C to allow refolding and secretion of the reporter protein. Cells were fixed at the indicated time points for IF or collected for EndoHf digestion. EndoHf was used according to the manufacturer's instructions. In brief, 10 μg WCE was incubated at 95 °C for 10 min to allow denaturing. Afterwards, the samples were incubated for 1 h at 32 °C with 1,000 U enzyme and separated for further analysis on SDS–PAGE. To analyse protein export by IF, cells transfected and treated as above were processed for IF 45 and 90 min post heat shock. To determine VSVG localization with respect to the Golgi, cells were stained with the *cis*-Golgi marker GM130. The level of secretion was defined according to the localization of VSVG in ER (pre-Golgi), Golgi or post-Golgi structures. Where indicated, a plasmid encoding dsRed fused to an ER retention signal was transfected.

**FACS and ELISA.** For secretion rate analysis by FACS, export assays were performed as described above using a ts045-VSVG-GFP variant expressing an internal FLAG-tag introduced in frame in the EcoNI site in the extracellular domain of the protein. Jsl1 cells were collected on shifting cells to 32 °C for 90 min and stained against the FLAG epitope for 30 min on ice in 3% bovine serum albumin in PBS. Cells were washed in PBS and stained with a Cy5 coupled α-mouse antibody for 30 min on ice in 3% bovine serum albumin in PBS. Cells were washed and resuspended in an adequate volume of PBS and analysed using a Guava easyCyte 8 FACS. Living cells were gated for GFP expression and analysed for FLAG surface staining.

Hek293 cells, WT and ΔE29, were seeded in 12-well plates and transfected with the VSVG variant described above. After the 40 °C heat shock, cells were trypsinized, kept at 32 °C for the indicated time, and then stained and analysed by FACS as described above.

IL2 sandwich ELISA was performed using the ELISA MAX Kit (BioLegend) according to the manufacturer's instructions. Primary human PBMCs were isolated as described[33] and T cells were enriched by removing adherent cells. Cells were then transfected with the *Sec16* E29 or control vivo-MO and stimulated with 1 μg precoated CD3 and 1 μg soluble CD28 for the indicated times. For total IL2 analysis, cells were treated with $5 \, \mu g \, ml^{-1}$ Brefeldin A (BFA) at 37 °C 40 min before collecting, washed in PBS and lysed in non-denaturing lysis buffer (20 mM Tris–HCl (pH 8.0), 137 mM NaCl, 10% glycerol, 1% NP40, 2 mM EDTA and proteinase inhibitors).

**Fluorescence microscopy and FRAP.** Cells were seeded on coverslips and transfected the following day. Fourty-eight hours later, cells were fixed in 4% paraformaldehyde, followed by equilibration with 0.1% Tx100/PBS and blocking in 5% goat serum/PBS. Primary antibody was incubated for 2 h at room temperature

and secondary antibody for 1 h at room temperature. After nucleus staining using Hoechst, the cells were mounted in Moviol and dried overnight before imaging. Jsl1 cells were collected and fixed in a 1.5 ml reaction tube followed by cytospin for 7 min at 200 r.p.m. Cells were then treated as described above. To study the impact of Sec16 CTR overexpression, HeLa cells were seeded on coverslips and transfected on the following day with plasmids encoding the four different Sec16 CTRs fused to a C-terminal GFP-tag. Fourty-eight hours post transfection, cells were fixed and stained with the *cis*-Golgi marker GM130. The Golgi morphology in untransfected cells appeared as one distinct structure close to the nucleus and was defined as 'intact'. Golgi stainings that showed a loose multimeric structure in the cell periphery were defined as a 'disrupted' Golgi. Dividing cells, as identified by Hoechst staining, were not considered as these cells showed a natural dispersed Golgi structure (see Supplementary Fig. 2 for examples).

Samples were imaged with either a Leica TC Sp2 or TC SP8 using a × 63 magnification oil immersion objective. Images were processed using ImageJ free software. For Sec16 and Sec31, quantification images were analysed using the ImageJ particle count tool. For this purpose, images were converted into 8 bit images in grayscale applying uniform thresholding. A scale was set to each picture and particles with a diameter above 60 nm were counted corresponding to the lower size limit of a COPII vesicle[29].

For FRAP experiments, cells were seeded in 4 cm Willco dishes in DMEM medium with 10% fetal bovine serum and transfected 1 day later with a Sec23-GFP-expressing plasmid. Twenty-four hours after transfection, medium was replaced by DMEM without carbonate and phenol red, and supplemented with 20 mM HEPES. Imaging was performed with a Leica SP2 equipped with a preheated 37 °C chamber using a × 63 oil immersion objective and the 488 nm Ar laser. Five prebleach images were acquired, followed by eight bleaching images with 100% laser intensity. After bleaching, 150 images were taken with a scanning speed of 400 Hz. FRAP recovery rate was determined after full-scale normalization, setting prebleach intensity to 1 and post-bleach intensity to 0. Calculation of half-life time was performed as previously described[43] using GraphPad Prism software.

**Statistical analysis.** The number of independent experiments is given in each figure. To test significance, Student's unpaired *T*-test was performed using standard office software. The following thresholds were used: *$P < 0.05$; **$P < 0.01$; ***$P < 0.001$.

**Data availability.** The authors declare that the data supporting the findings of this study are either available within the article (and its Supplementary Information files) or available on request by emailing the corresponding author.

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

## Acknowledgements

We thank David Stephens (Bristol, UK) for providing a GFP-Sec16 expression construct and the Sec24C antibody. Stefan Mundlos (Berlin, Germany) provided CRISPR/Cas9 plasmids. Sophie Veitinger (Marburg, Germany) initially helped with microscopy, Monika Michel (Marburg, Germany) provided technical assistance and Lara Kämmerer contributed as a rotation student. Funding was provided by an Emmy-Noether-Fellowship of the DFG to F.He. (HE 5398/3); additional support came from the DFG-funded SFB958/A21.

## Author contributions

I.W. performed most of the experiments and was supported by M.P., A.N. and O.H. R.K. generated genome-engineered cell lines and performed initial phenotyping. R.J. and F.Ho. contributed essential equipment and expertize in microscopy studies, especially FRAP. F.He. and I.W. planned the experiments, analysed the data and wrote the manuscript. F.He. initiated and designed the study, and supervised the work.

## Additional information

**Competing financial interests:** The authors declare no conflict of interests.

