## [Peer Review File · Nature Communications]

Reviewers' comments:

Reviewer #1 (Remarks to the Author):

I reviewed an earlier version of this manuscript for another journal. At the time, I was supportive but had a number of suggestions for improvement. Those suggestions have been addressed to my satisfaction in this version, and I have no further requests (except that "endoplasmatic" in the Abstract should be "endoplasmic").

Here is a summary of what the paper shows and why I consider it significant.

=====

This study characterizes alternative splicing isoforms of the major Sec16 protein in mammalian cells. It is shown that upon T cell activation, Sec16 splicing is altered, presumably to help accommodate the increased cargo load in the secretory pathway. Alternative splicing alters the C-terminal domain, which is the "business end" of the protein with regard to interaction with Sec23 and other COPII components. In particular, an isoform containing exon 29 but not exon 30 shows increased binding to Sec23, and overexpression of a C-terminal domain containing only exon 29 potently disrupts Golgi structure. This disruption can be overcome by concomitantly overexpressing Sec23, providing evidence that the overexpressed C-terminal domain is titrating out Sec23. Moreover, genome engineering to remove exon 29 reduces the number of ER exit sites (ERES).

The novelty of this paper lies in the evidence that alternative splicing plays a role in regulating secretion at the level of COPII. Little attention has been paid to this potential regulatory mechanism. In addition, the finding that overexpression of Sec23 suppresses the effects of overexpressing the C-terminal domain is a nice contribution, as is the analysis of how alternative splicing affects binding of the C-terminal domain to COPII components. In general, the experiments are executed and presented well.

Although a detailed mechanistic interpretation is precluded by our limited understanding of the functions of Sec16, this study offers unique insights. I believe that it will be viewed as an important contribution and will stimulate further research on the role of alternative splicing in regulating the secretory pathway.

Reviewer #2 (Remarks to the Author):

Sec16 undergoes alternative splicing regulation after human T cell stimulation (Martinez et al., 2012). The authors start their work by confirming the reported changes in the Sec16 splicing pattern after PMA stimulation. Activated T cells exhibit increased inclusion of exon 29 (E29), with concomitant decrease of full-length and exon 30-containing isoforms. Sec16 alternative splicing modifies its C-terminal domain, which interacts with Sec12 and Sec23, and is essential to mediated COPII transport. Previous work demonstrated a dominant negative effect of Sec16 C-terminal

domain overexpression in Golgi morphology (Martinez et al., 2012). Now the authors show that cells overexpressing the E29-only C-terminal domain cause a more dramatic effect in ER to Golgi trafficking. Upon stimulation, T cell increase the rate of ER-export as measured by Sec31 and Sec24 COPII-positive structures and reporter assays. They developed a morpholino antisense oligo that interferes with E29 inclusion, confirm its activity on splicing and show that it impairs the PMA-induced increase of ERES and export efficiency. Lastly, knock out of E29 by CRISPR/CAS9 negatively affects COPII-mediated trafficking and recapitulates the effects of the morpholino. Since immunoprecipitation assays with recombinant C-terminal domains of Sec16 showed a role of E29 in the interaction with Sec23, the authors suggest a model in which alternative splicing of Sec16 regulates the increase of COPII-mediated trafficking upon T cell activation though enhanced recruitment of Sec23 and increased ER-Golgi trafficking.

Major points:

Figure 1d: The shift observed on the NU-PAGE gel could be also due to other kind of modification (such as phosphorylation). To confirm this result the authors should perform a NU-PAGE gel of PMA induced T cell transfected or not with Morpholino regulating Sec16 splicing.

Figure 2: The authors should also show the representative images of the other variants of C-terminal domain. Moreover, how do the authors explain that short isoform behaves as FL domain (see also comment in Fig.4-7)? Can they rule out a regulatory role for E30?

Figure 4-6,7: the authors claim that "morpholino completely reverts splicing of E29" (lane 187; Fig4b). However, the morpholino also increases the expression of the short isoform, lacking both exons. So it does not revert the splicing of Sec16 to control levels and the role of the additional variants should be investigated to rule out their contribution to the phenotype. For instance, in Fig.6a, it is shown that the Sec16 "short" isoform does not interact with the earliest component of COPII-mediated trafficking (Sar1 proteins). Therefore, how can the authors be sure that the observed effects in Fig. 4e-g, are due to the lack of E29 (and interaction with Sec23) and not to the reduced interaction of Sec16 with Sar1 proteins?

Figure 4d: Since the FL and Δ 29 Sec16 isoforms in SDS-PAGE cannot be distinguished, Nu-PAGE gel should be performed to assess the functional relevance of the splicing switch induced by morpholino at the protein level. Moreover, if the full length protein decreases upon PMA stimulation, why is the higher band increased? Which variant corresponds to which band?

Minor points:

Figure 1a: The authors should depict more precisely the structure and alternative splicing of Sec16 gene.

Line 112-115: The authors should explain better the aim of CHX experiment in supplementary fig.1c.

Reviewer 2:

We are happy about the overall positive response and thank the reviewer for pointing out some very interesting questions. We have addressed these points as follows:

Figure 1d: The shift observed on the NU-PAGE gel could be also due to other kind of modification (such as phosphorylation). To confirm this result the authors should perform a NU-PAGE gel of PMA induced T cell transfected or not with Morpholino regulating Sec16 splicing.

We have performed the suggested experiment and have included a NU-PAGE gel confirming altered Sec16 isoform expression in stimulated T cells upon MO transfection (Fig. 4d). The result is consistent with the slower migrating isoform being the exon 29 isoform and the faster migrating one the short isoform (discussed in the text). This is also consistent with the RT-PCRs showing that these two isoforms are the major isoforms in T cells.

Figure 2: The authors should also show the representative images of the other variants of C-terminal domain. Moreover, how do the authors explain that short isoform behaves as FL domain (see also comment in Fig.4-7)? Can they rule out a regulatory role for E30?

We have included additional images of the other C-terminal variants as Supplementary Figure 2b. The question whether other isoforms, e.g. E30, may also play a regulatory role is very interesting and we have performed another MO experiment to experimentally address this point. We have used a MO against exon 30 and measured Sec16 isoform expression as well as the effect on ERES number upon stimulation. This MO leads only to a slight (and not significant) reduction of ERES number compared to the CoMO, despite a strong effect on the FL and the E30 isoforms (Supplementary Figure 7). Furthermore, combination of MOs targeting exons 29 and 30 does not increase the effect on ERES number compared to the E29MO alone (Supplementary Figure 7b).

These experiments make a predominant role of the FL and E30 isoforms very unlikely. However, the E29MO also leads to a strong increase in the short isoform which is only partly recapitulated by the E30MO. We can therefore not formally rule out that parts of the effect observed in the E29MO treated cells are due to an increase in the short isoform. However, all our data, from the observation in Fig. 1 that the E29 isoform is strongly increased during activation whereas the short isoform is unchanged or slightly reduced, to the experiments in Fig. 2 showing that the E29 isoform has the strongest impact on ER-to-Golgi transport and the combined Morpholino and CRISPR/Cas9 experiments, point to a predominant role of the E29 isoform. We therefore think that our model is valid and well supported by our data. However, we have also added a more detailed discussion pointing out that 'it appears possible that expression of the other Sec16 splice variants and ultimately the Sec16 isoform ratio contribute to the regulation of ER export efficiency in the endogenous situation'.

Figure 4-6,7: the authors claim that "morpholino completely reverts splicing of E29" (lane 187; Fig4b). However, the morpholino also increases the expression of the short isoform, lacking both exons. So it does not revert the splicing of Sec16 to control levels and the role of the additional variants should be investigated to rule out their contribution to the phenotype. For instance, in Fig.6a, it is shown that the Sec16 "short" isoform does not interact with the earliest component of COPII-mediated trafficking (Sar1 proteins). Therefore, how can the authors be sure that the observed effects in Fig. 4e-g, are due to the lack of E29 (and interaction with Sec23) and not to the reduced interaction of Sec16 with Sar1 proteins?

Please see above for the E30MO experiment and the additional discussion added to this point. We fully agree that the differential interaction of Sar1 with Sec16 isoforms is very interesting and we discuss that Sec16 isoforms may differentially control Sar1 activity. We have actually started to generate recombinant proteins to address this point experimentally. However, this is an entirely new project based on the findings of the present manuscript. Addressing this question requires a substantial amount of work, which we think is far beyond the scope of this initial report.

Figure 4d: Since the FL and $\Delta 29$ Sec16 isoforms in SDS-PAGE cannot be distinguished, Nu-PAGE gel should be performed to assess the functional relevance of the splicing switch induced by morpholino at the protein level. Moreover, if the full length protein decreases upon PMA stimulation, why is the higher band increased? Which variant corresponds to which band?

Please see above for the MO-Nu-PAGE experiment. We also discuss the identity of the two bands observed in the Nu-PAGE gels (Page 7).

Figure 1a: The authors should depict more precisely the structure and alternative splicing of Sec16 gene. Line 112-115: The authors should explain better the aim of CHX experiment in supplementary fig.1c.

We have added a scheme showing alternative splicing of Sec16 (Fig. 1a). We have also added a more detailed explanation for the CHX experiments. We have also included more data points in the FRAP experiment; this does not alter the conclusion.

REVIEWERS' COMMENTS:

Reviewer #2 (Remarks to the Author):

In this revised manuscript, the authors have addressed most of the criticisms raised to the original work. The manuscript appears improved.